# On developing a new ionospheric plasma index for the Brazilian equatorial F region irregularities

Laysa Cristina Araujo Resende[1,2], Clezio Marcos Denardini[1], Giorgio Arlan Silva Picanço[1], Juliano Moro[2,3], Diego Barros[1], Cosme Alexandre Oliveira Barros Figueiredo[1], Régia Pereira Silva[1].

[1]National Institute for Space Research (INPE) S. J. Campos, SP, Brazil
[2]National Space Science Center, China Academy of Science, CAS, Beijing, China
[3]Southern Regional Space Research Center – CRS/INPE, Santa Maria, RS, Brazil

*Correspondence to*: Laysa C. A. Resende (laysa.resende@inpe.br; laysa.resende@gmail.com)

**Abstract.** F region vertical drifts ($V_z$) are the result of the interaction between the ionospheric plasma with the zonal electric field and the Earth's magnetic field. Abrupt variations in $V_z$ are strongly associated with the occurrence of plasma irregularities (spread-F) during the nighttime periods. These irregularities are manifestations of the space weather in the ionosphere environment without necessarily require a solar burst. In this context, the Brazilian Space Weather Study and Monitoring Program (Embrace) of the National Institute for Space Research (INPE) has been developing different indexes to analyze these ionospheric irregularities in the Brazilian sector. Therefore, the main purpose of this work is to produce a new ionospheric scale based on the analysis of the ionospheric plasma drift velocity, named AV. It is based in the maximum value of $V_z$ ($V_{zp}$), which in turn is calculated through its relationship with the virtual height parameter, *h'*F, measured by the Digisonde Portable Sounder (DPS-4D) installed in São Luís (2°S, 44° W, dip: -2,3°). This index quantifies the time relation between the $V_z$ peak and the irregularity observed in the ionograms. Thus, in this study, we analyzed 8 years of data, between 2009 and 2015, divided by season in order to construct a standardized scale. The results show there is a delay of at least 15 minutes between $V_{zp}$ observation and the irregularity occurrence. Finally, we believe that this proposed index allows evaluating the impacts of ionospheric phenomena in the space weather environment.

# 1 Introduction

Ionospheric irregularities or spread-F occur in the F region, which are characterized by regions of signal scattering in ionosondes. In general, the spread-F may be associated with plasma bubbles characterized by regions where the plasma density is reduced. Also, these irregularities usually develop after sunset (Abdu et al., 1983; Abdu, 2001). The plasma bubbles are generated through the nonlinear evolution of the Rayleigh-Taylor instability in equatorial regions (Bittencourt and Abdu, 1981; Abdu et al., 2006; Huang et al., 2002; Abdu et al. 2009a).

One of the most useful parameter to analyze these irregularities is the vertical drift velocity ($V_z$), which is a response of the zonal electric field in the F region, and it is controlled by the interaction between the E- and F-layers, being positive (upward) during the day. In the nighttime period, the $V_z$ become negative due to the inversion of neutral wind (Abdu et al., 2006). Soon before the inversion, an increase in plasma drift occurs lifting the equatorial F layer and controlling the generation of plasma bubbles (Fejer et al., 1991, 2008, Huang et al., 2002). This phenomenon is named pre-reversal enhancement (PRE) of the vertical plasma drift and gives rise to a maximum in the vertical velocity drift ($V_{zp}$) around 18:00-19:00 LT (Heelis et al., 1974; Farley et al., 1986).

It is well established that the PRE presents a great variability in relation to seasonality, solar cycle and magnetic activity (Fejer, 1991, Abdu et al., 1995, Fejer et al., 2008). In the Brazilian sector, the ionospheric irregularities occur frequently in summer due to the relatively small angle between the solar terminator and the magnetic meridian (Batista et al., 1986). Therefore, there is an almost instantaneous decoupling between the E and F regions. Thus, the polarization electric fields of the F region associated with the PRE peak have higher amplitudes and, the $V_z$ reaches higher velocities.

Regarding the solar flux, some works have pointed out a direct correlation between $V_z$ and F10.7 radio index (Fejer et al., 1979, Fejer et al.,1991, Batista et al.,1996). In fact, the number of free electrons in the ionosphere will increase with the solar flux intensification causing intense electric fields. Consequently, larger amplitudes of the $V_z$ parameter are observed.

Although several studies report the plasma irregularities occurrences in the F region with plasma $V_z$ increases, there is still no climatological study that related the cause and effect of the solar-terrestrial applied to the ionosphere to construct an ionospheric index. Therefore, there are no indexes related to the

irregularities in ionospheric plasma as spread-F (plasma bubbles). Jakowski et al. (2012) suggested a disturbance ionosphere index (DIX), in which it describes the perturbation degree of the ionosphere using the Global Navigation Satellite Systems (GNSS) data. Recently, Nishioka et al. (2017) related the plasma irregularity occurrence using a index based on the Total Electron Content (TEC) measurements. However, an index which correlates to the $V_z$ with the irregularity/plasma bubbles occurrences was not found in the literature.

Thus, in this study we present the new developed ionospheric index, AV, based on the $V_z$ parameter. This index quantifies the time relation between the $V_z$ peak, and the irregularity observed in the ionograms. The time relation between these parameters ($V_{zp}$ and irregularity observation) is at least 15 minutes for values of $V_{zp}$ less than 60 m/s. Finally, this study demonstrated that the AV index can be used for space weather forecast, and it will help in the evaluation of the phenomena impacts in the ionosphere.

**2 Data Set**

We have used ionospheric data in the Brazilian sector, São Luís (2º 31' S, 44º 16' W, dip: -2.3°). The data were acquired by a Digisonde Portable Sounder (DPS4), an ionospheric radar that operates in variable high frequency (HF). The data are composed of the signal reflected by the ionospheric layers, in which they are registered in ionograms, graphs of frequency versus virtual height (*h'*F). Therefore, it is possible to calculate the electron density profile and parameters of the different regions in the ionosphere (Reinisch, 1986; 2009).

The $V_z$ parameter is a representation of the vertical drift velocity of the post-sunset F-region (Bittencourt and Abdu, 1981, Abdu et al., 1983, Abdu et al., 2006). It is important to mention that the heights below 300 km were not considered in this study since the ionosphere plasma is subject to the recombination effects. We have calculated the $V_z$ using its relation with *h'F* determined as $Vz = \dfrac{\Delta h'F}{\Delta t}$.

The *h'*F parameter is collected every 15 minutes interval on a continuous basis. We have used *h'*F values over 18:00-21:00 UT (21:00-24:00 LT) of each night to obtain the $V_z$ parameter for the present analysis.

The higher vertical drift velocity in each day, or $V_{zp}$, is considered to represent the PRE. As the purpose of this analysis is to quantify the time relation between $V_{zp}$ and the ionospheric irregularity occurrences, we firstly analyzed two years representing high (year 2001) and low (descending phase) (year 2015) solar flux. It is important to mention that the criterion was to choose two whole years in different phases of the solar cycle that had a complete and revised amount of data to avoid any error in the results. We used this analysis to construct the AV index scale. In the following step, we performed a climatological study considering the data acquired from the years 2009 to 2014 in order to validate the developed AV scale. Lastly, the dataset was separated in seasons: equinoxes (March, April and May, September, and October), summer (November, December, and January), and winter (June, July, and August), since the irregularities are observed every night between November and January. In other words, since plasma bubbles occur between September and March, we eliminate the two months before and after this analysis. The same criterion was used in the winter season.

## 3 Results and Discussions

### 3.1 AV Index Scale

The purpose of this analysis is to quantify the interrelationship between $V_{zp}$ and the ionospheric irregularity occurrences that may be associated with plasma bubbles (spread-F). Based on previous studies about this correlation (Batista et al., 1986, Batista et al., 1996, Abdu et al., 2001), it can be assumed that the lowest and highest variations in the $V_{zp}$ amplitude are, approximately, 20 and 70 m/s, respectively. Therefore, we analyzed the irregularity occurrence after the highest $V_{zp}$ value, and we proposed a scale to quantify this relationship.

The new ionospheric scale AV is divided into five levels as shown in Table 1. The colors indicate the level of disturbance in increasing order of magnitude. $AV_1$ and $AV_2$ (blue/green) imply the typical conditions, when no irregularities in the ionospheric plasma have been observed. From the $AV_3$ (yellow) and above it is possible to observe the spread in the F region detected by the Digisonde. The $AV_4$ and $AV_5$ indexes, represented respectively by orange and red colors, meaning extreme conditions, which the existence of plasma bubbles are more probable. The color selection was used based on the Embrace index development program.

Table 1: The new ionospheric plasma index, AV, divided into five levels.

The colors indicate the level of disturbance. $V_{zp}$ is given in m/s.

| AV$_1$ | $20 \leq V_{zp} < 30$ |
|--------|------------------------|
| AV$_2$ | $30 \leq V_{zp} < 40$ |
| AV$_3$ | $40 \leq V_{zp} < 50$ |
| AV$_4$ | $50 \leq V_{zp} < 60$ |
| AV$_5$ | $V_{zp} \geq 60$       |

It has been known from previous studies that irregularities in F region can be observed when $V_{zp}$ amplitudes are higher than 30 m/s (Abdu et al., 1985; Fejer et al., 1999). In the Brazilian region, Abdu et al. (1983) and Abdu et al. (2009a) found that the $V_{zp}$ should be around 30-50 m/s to observe the irregularity/spread-F. Other authors found a threshold of 40 m/s in different locations (Basu et al., 1996; Whalen, 2003). Furthermore, the irregularity occurrence probability becomes higher when the $V_{zp}$ is greater than 40 m/s (Huang et al., 2015). In fact, we do not observe in our analysis an expressive irregularity occurrence for $V_z$ less than 40 m/s. Therefore, the threshold was selected as 40 m/s here to validate the proposed AV index. In other words, the values smaller than 40 m/s refer that the irregularity will not appear (AV$_1$) or will rarely appear (AV$_2$) since the interest of this study is related to general cases.

Since the $V_{zp}$ intensification may indicate the occurrence of plasma irregularities in ionograms, mainly plasma bubbles, we performed a statistical analysis to find the correlation between the instant in which $V_{zp}$ increases with the time that the irregularity is identified. An example is shown in Table 2 for January 11, 2001. It is observed that $V_{zp}$ reaches 43 m/s at 21:45 UT that corresponds to the AV$_3$ index. The irregularity can be observed in the ionogram 45 minutes after the $V_{zp}$ peak (22:30 UT), as shown in Figure 1. The red arrows indicate the spread in the F region.

Table 2 -Example of the relation between the irregularity occurrence and $V_{zp}$ parameter.

| Day | $V_{zp}$ | Index Scale | Time | Spread F Time Occurrence |
|---|---|---|---|---|
| 01-11-2001 | 43 m/s | $AV_3$ | 21:45 UT | 22:30 UT |

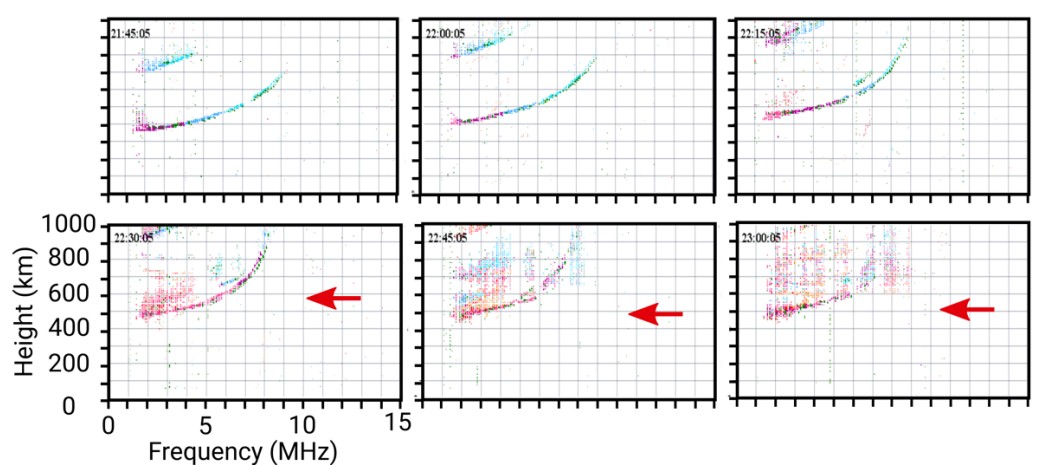

**FIGURE 1** – Sequences of ionograms showing the spread-F over São Luís on January 11, 2001 (red arrows).

## 3.2 AV Index Validation

Table 3 presents the statistics of the data used in the analysis for 2001 and 2015, considering the total number of observations, number of days in which the irregularities were observed, and the number of the days that their classification was the $AV_3$, $AV_4$ or $AV_5$.

Table 3 – Statistics of the data used in this study for 2001 and 2015.

| Year | Season | Num. of observations | Num. of irregularities | Index | Num. of Days per index |
|---|---|---|---|---|---|
| 2001 | Summer | 82 | 66 | $AV_3$ | 36 |
| | | | | $AV_4$ | 25 |
| | | | | $AV_5$ | 5 |
| | Equinoxes | 167 | 63 | $AV_3$ | 41 |
| | | | | $AV_4$ | 21 |
| | | | | $AV_5$ | 1 |

| Year | Season | | | Index | Value |
|---|---|---|---|---|---|
| | Winter | 84 | 61 | AV$_3$ | 10 |
| | | | | AV$_4$ | 0 |
| | | | | AV$_5$ | 0 |
| **2015** | Summer | 84 | 61 | AV$_3$ | 29 |
| | | | | AV$_4$ | 23 |
| | | | | AV$_5$ | 9 |
| | Equinoxes | 153 | 58 | AV$_3$ | 35 |
| | | | | AV$_4$ | 21 |
| | | | | AV$_5$ | 2 |
| | Winter | 70 | 23 | AV$_3$ | 23 |
| | | | | AV$_4$ | 0 |
| | | | | AV$_5$ | 0 |

Figure 2 shows the statistical analysis between the $V_{zp}$ values considering only the indexes AV$_3$, AV$_4$ and AV$_5$ and the time that the irregularity starts to appear in the ionogram. Firstly, the analysis takes into account the data gathered in the summer of 2001 and 2015 over São Luís. The interval between the $V_{zp}$ peak and the observation of the spread-F was discretized into 5 intervals for the sake of the analysis: 0, 15, 30, 45, and ≥60 min. This quantity is called $\Delta t_{vi}$ hereafter. Notice that this granularity of 15 minutes is a limitation of the Digisonde sampling time and all the occurrences in which $\Delta t_{vi}$ is higher than 60 minutes were placed on the last interval.

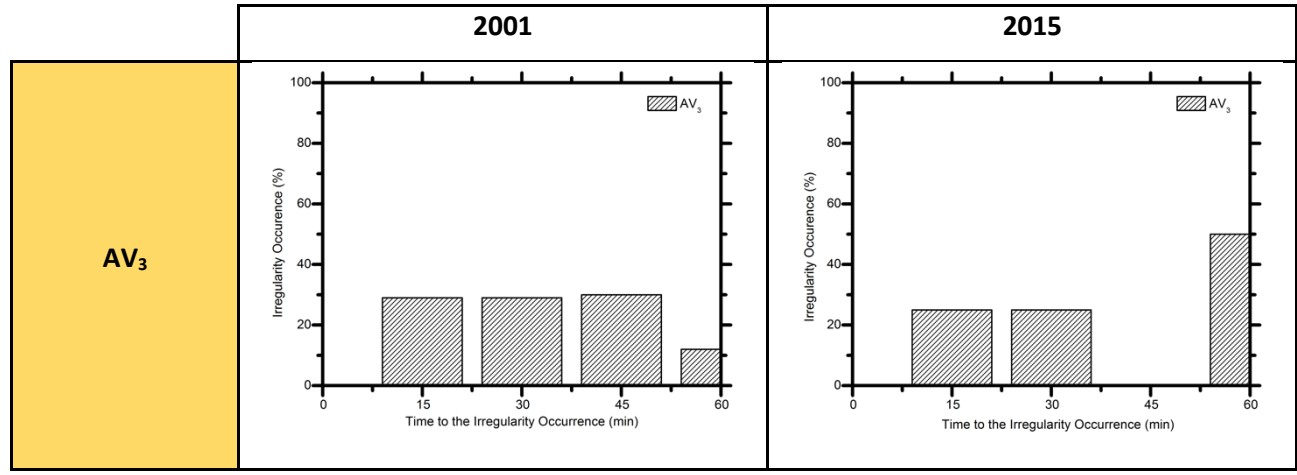

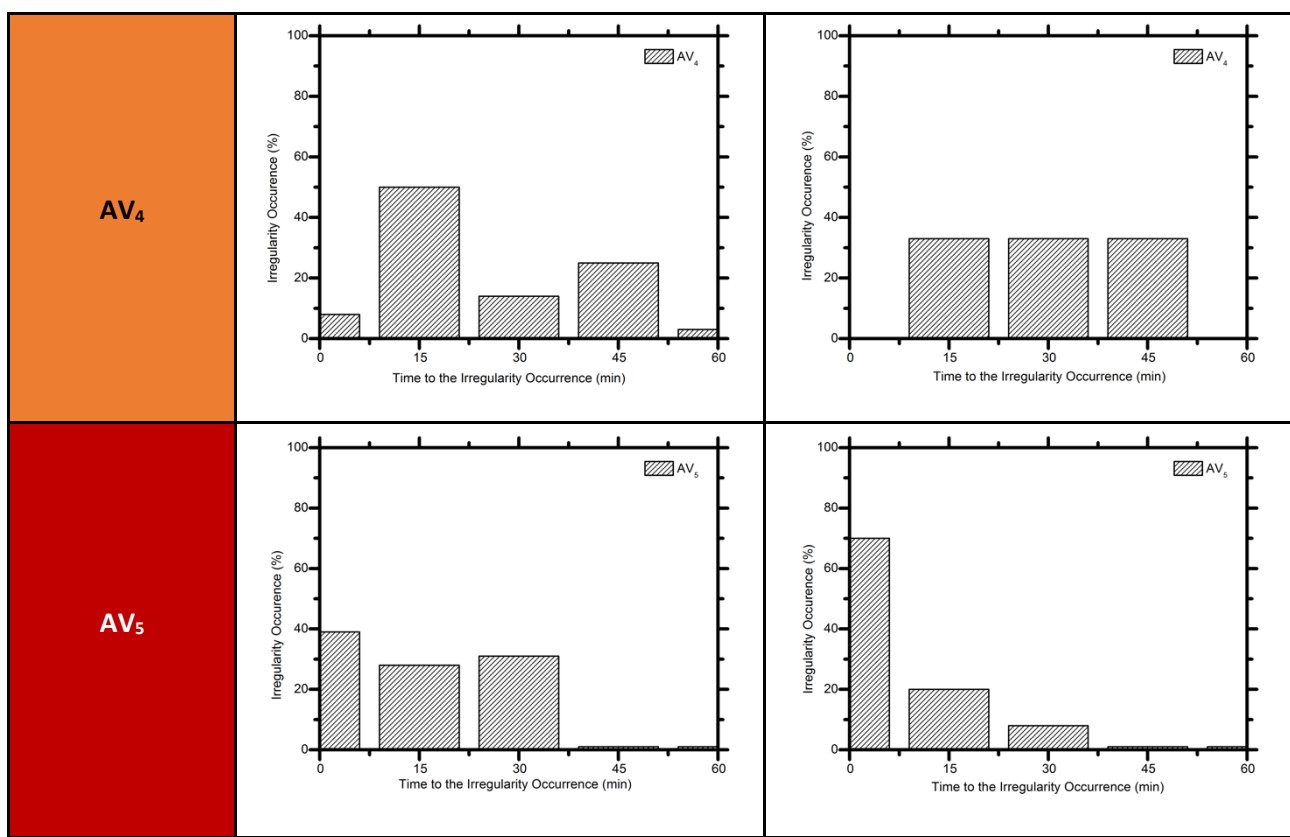

**FIGURE 2** – Time relation between the $V_{zp}$ parameter, considering $AV_3$, $AV_4$ and $AV_5$ indexes, and the irregularity observations in ionograms in the summer of 2001 and 2015 over São Luís.

Comparing the results for the $AV_3$ index, it is observed that $\Delta t_{vi}$ is at least 15 minutes in both solar cycles. In 2015, no data with $\Delta t_{vi}$ equal 45 minutes was observed. There was also a significant change comparing both scenarios with respect to the occurrences with $\Delta t_{vi}$ equal to or greater than 60 minutes. In 2001, only 16% of the cases fell in this interval whereas in 2015 it was 50%. The high occurrence in the descending phase of the solar cycle could be related to the irregularities post-midnight occurrence and, generally, they are observed in low solar activity (Otsuka, 2018).

Regarding the $AV_4$ index, 10% of the cases had a $\Delta t_{vi}$ equal to 0 minute during the maximum solar flux (2001), meaning that the spread-F occurred at the same time of the $V_{zp}$. On the other hand, in 2015 no data fell into this interval. However, notice that there is a high probability that $\Delta t_{vi}$ is equal to or higher than 15 minutes.

When the index reached the $AV_5$ corresponding to extreme cases, we observe a high occurrence of events with $\Delta t_{vi}$ equals 0 minute. In fact, notice that almost 40% of that data in 2001 and 70% of the data in 2015 comprise this interval.

It can be inferred from our results that the probability density function of $\Delta t_{vi}$ for $AV_3$ and $AV_4$ approaches a uniform distribution with lower bound at 15 minutes, except for 2001 and $AV_4$. Regarding $AV_5$ in 2001, the distribution seems uniform with lower bound at 0 minute and upper bound at 30 minutes. For the same index, the distribution appears to be exponential in 2015. Furthermore, it can be concluded that the probability to observe $\Delta t_{vi}$ less than 15 minutes given that the index is $AV_3$ or $AV_4$ is negligible.

It is well established that the spread-F occurrence depends on the season and epoch in the solar cycle (Abdu et al., 1983; Abdu et al., 1995; Abdu, 2001). Abdu et al. (1985) showed that the drift velocities were small during the low sunspot years, which weaken the irregularities development. Huang et al. (2002) observed that the maximum irregularities rates were significantly higher during the maximum phase of the solar cycle. This behavior happens because the thermospheric winds and longitudinal gradients in conjugate E layer conductivity are more effective. Those are the key parameters that control the evening F region dynamo electric field. Therefore, during the high solar activity, there are significant variations in thermospheric winds and in longitudinal conductivity gradients of the evening conjugate E layers, which lead to higher values of $V_{zp}$ and, consequently, a favorable environment for the irregularity formation (Abdu et al., 1983).

In our analysis, we observed a significantly high number of irregularities in 2015. We believe that it is caused due to the descending phase of the solar cycle. However, although the irregularities reached the $AV_5$ level in 2015, its duration in ionograms was lower than that in 2001. This can be seen in Figure 3, where we show the duration of spread-F in ionograms for $AV_3$ level in 2001 and 2015. The duration of irregularity was divided into 5 intervals: less than 6 hours (t<6), between 6 and 7, 7 and 8, 8 and 9 (6<t<7; 7<t<8; 8<t<9), and ≥ 9 hours. The difference between the years is very clear, which in high solar cycle the duration of irregularities is more than 9 hours in most of the events. On the other hand, in 2015 the spread-F last less than 6 hours almost 70% of cases, revealing a solar flux influence and agreeing with the previous study (Abdu et al., 1985; Huang et al., 2002).

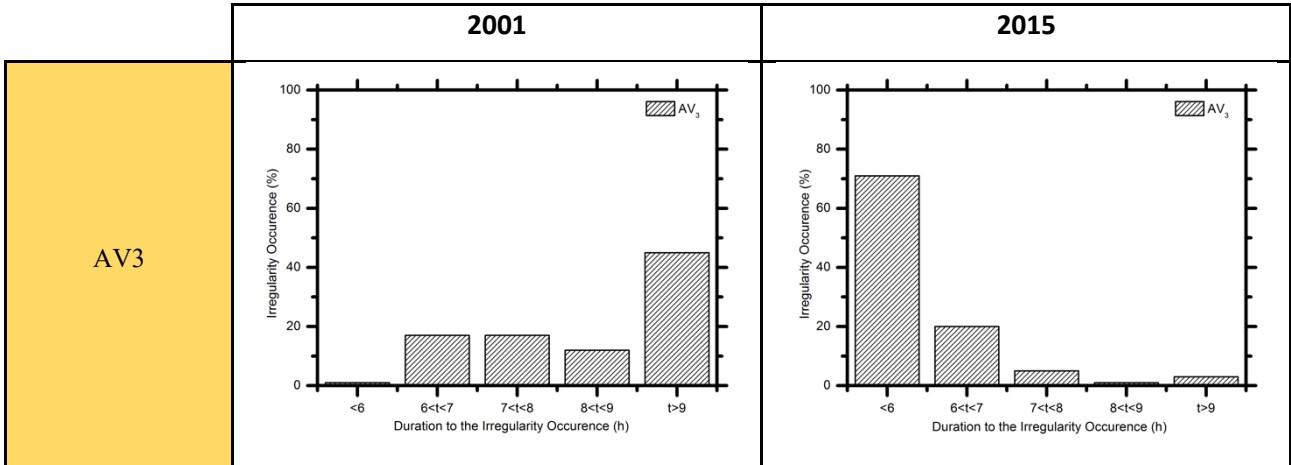

**FIGURE 3** – Occurrence of spread-F in ionograms for $AV_3$ index in 2001 and 2015 over São Luís to evidence the differences between the solar flux levels.

In the early hours, the irregularities were controlled by the polarization electric field, but after 6 hours it is the ionospheric plasma dynamics that control the bubble (Huang et al., 2011). Barros et al. (2018) show an important role of the zonal wind in the evolution of the plasma bubbles in the TEC data between 2012 and 2016. They did not discuss the differences in the solar cycle phases. However, they showed a good agreement between the zonal drift velocities and the thermospheric winds in the plasma bubbles occurrence. Therefore, we believe that the thermospheric winds in 2015 were lower than 2001 interfering in the growth rate of Rayleigh-Taylor instability (discussed below). This fact will be evaluated in greater detail in future work.

As mentioned before, the spread-F is directly related to the plasma bubbles. These plasma bubbles are formed due to the Rayleigh-Taylor gravitational instability process which is operational on the steep upward gradient in the nighttime bottom side of the F region at the magnetic equator (Abdu et al., 2006). This Rayleigh-Taylor mechanism is related to the linear growth rate ($\gamma$) given by (Haerendel et al., 1992; Sultan, 1996; Abdu et al., 2001; Abdu et al., 2006)

$$\gamma = \frac{\sum_F P}{\sum_E P + \sum_F P} \left( \frac{E}{B} - U^P + \frac{g}{\nu} \right) \frac{1}{L} - \beta, \qquad (1)$$

where the $\sum P$ is the Pedersen conductivity in the field-line integrated for the E and F; $U^P$ is the Pedersen conductivity-weighted neutral wind perpendicular to the magnetic field in the magnetic meridian plane; $E$ and $B$ are the ambient zonal electric field and the magnetic field intensity, g is the gravitational acceleration, $\nu$ is the collision frequency, $L$ is the MacIlwain parameter, and $\beta$ is the recombination loss rate. All the terms in this equation were discussed in Abdu (2001). However, we highlight here that the angle formed between the solar terminator and the magnetic meridian is relatively small in summer, and consequently there is an almost instantaneous decoupling between the E and F regions (Batista et al.,1986). Thus, the polarization electric fields of the F region associated with the $V_z$ peak have higher amplitudes, favoring the instability growth. Thus, the irregularity is most favored to occur in summer than winter (Tsunoda, 1985, Barros et al., 2018).

In order to investigate the seasonal behavior, we analyzed the AV index for equinoxes and winter in 2001 and 2015 over São Luís. The results are presented in Figures 4 and 5 for the equinoxes and winter, respectively.

In relation to equinoxes (Figure 4), the $V_{zp}$ did not reach $AV_5$ in both years. For $AV_4$, only a few cases were observed in the high solar cycle. Among those, 10% had $\Delta t_{vi}$ equal 0 minute, 34% had $\Delta t_{vi}$ equal 15 minutes, 15% had $\Delta t_{vi}$ equal 30 minutes, 10% had $\Delta t_{vi}$ equal 45 minutes, and 31% had $\Delta t_{vi}$ equal to or greater than 60 minutes. Regarding $AV_3$, the irregularities were observed between 15 and 45 minutes after $V_{zp}$ peak in both years. No significant values with $\Delta t_{vi}$ equal 0 minute or greater than 60 minutes were found.

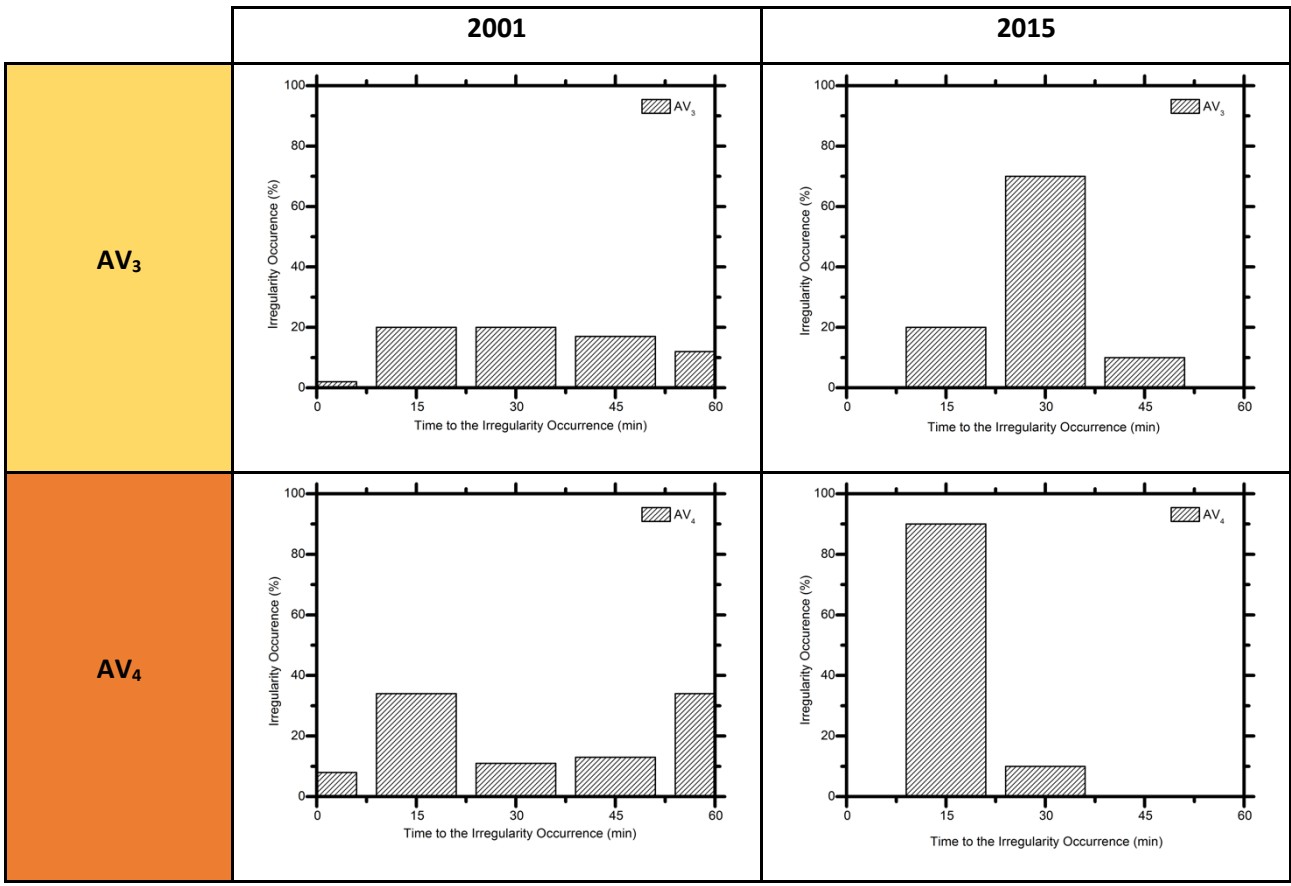

**FIGURE 4** – Time relation between the $V_{zp}$ parameter, considering AV3, and AV4 indexes, and the irregularity observations in ionograms in the equinoxes of 2001 and 2015 over São Luís. The $V_{zp}$ did not reach the AV5 scale.

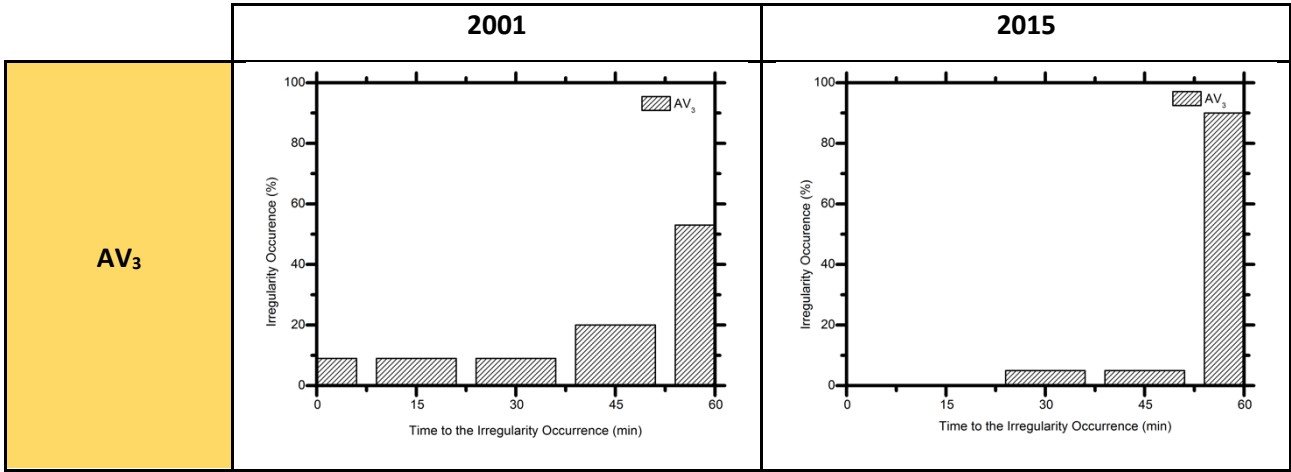

**FIGURE 5** – Time relation between the $V_{zp}$ parameter, considering $AV_3$ index, and the irregularity observations in ionograms in the winter of 2001 and 2015 over São Luís. The $V_{zp}$ did not reach the $AV_4$ and $AV_5$ scales.

As seen in Figure 5, the index did not reach $AV_4$ or $AV_5$ in winter. In high solar cycle, 10% of the cases had $\Delta t_{vi}$ equal 0 minute, 10% had $\Delta t_{vi}$ equal 15 minutes, 10% had $\Delta t_{vi}$ equal 30 minutes, and almost 20% had $\Delta t_{vi}$ equal 45 minutes. Thus, most of the irregularity observations occurred after one hour from the $V_{zp}$ peak. In general, the irregularities observed in the ionograms had $\Delta t_{vi}$ equal or greater 60 minutes in 2015.

Therefore, we observed only a few cases of spread-F in both seasons, agreeing with previous studies that irregularities related to plasma bubbles are more frequent in summer (Barros et al., 2018). Thus, we considered the analyze of the summer is enough to validate the proposed AV scale. However, it is important to mention that the index is valid for the other stations of the year too, as shown in the examples above. The climatological results are presented in the following section.

### 3.3 Climatological study between AV index and spread-F in summer.

Figure 6 shows the relation between $V_{zp}$ intensification, AV index, and the time that the irregularity starts to appear in ionogram. This study considered the data obtained from São Luís station during the summer of 2009 up to 2014. The years 2009 and 2010 refer to the minimum solar cycle 24,

2011 and 2012 were related to the ascending phase of the same solar cycle, and 2013 and 2014 to its maximum. In this figure, we also show the quantity of available observations and the number of days that we used in the analysis (below the year). Also, we show the quantity for each scale in the AV index used in this analysis.

It is possible to observe from the results that there is no regular pattern between the highest $V_{zp}$ value and the irregularity observations. However, it is important to mention that the probability to observe $\Delta t_{vi}$ equal to or higher than 15 minutes is still very high in all phases of the solar cycle.

After the $AV_3$ and $AV_4$ results, we did not observe significant values with $\Delta t_{vi}$ equal 0 minute. The most of events were found that $\Delta t_{vi}$ lies between 30 minutes and greater than 60 minutes, showing the

same pattern of the results presented in the previous section. Additionally, a high number of observations with $\Delta t_{vi}$ greater than 60 minutes were found, mainly in 2009, 2010, and 2014. As mentioned before, this percentage can be associated with the irregularities that occurred in post-midnight hours, which is a common observation in the low solar activity. Otsuka (2018) affirms that during solar minimum conditions, post-midnight irregularities may occur mostly in association with plasma bubbles initiated

around midnight. In addition, the the post-midnight plasma bubbles could be caused by atmospheric gravity wave seeding of the Rayleigh-Taylor instability and/or increase in growth rate of the Rayleigh-Taylor instability due to the F layer uplift.

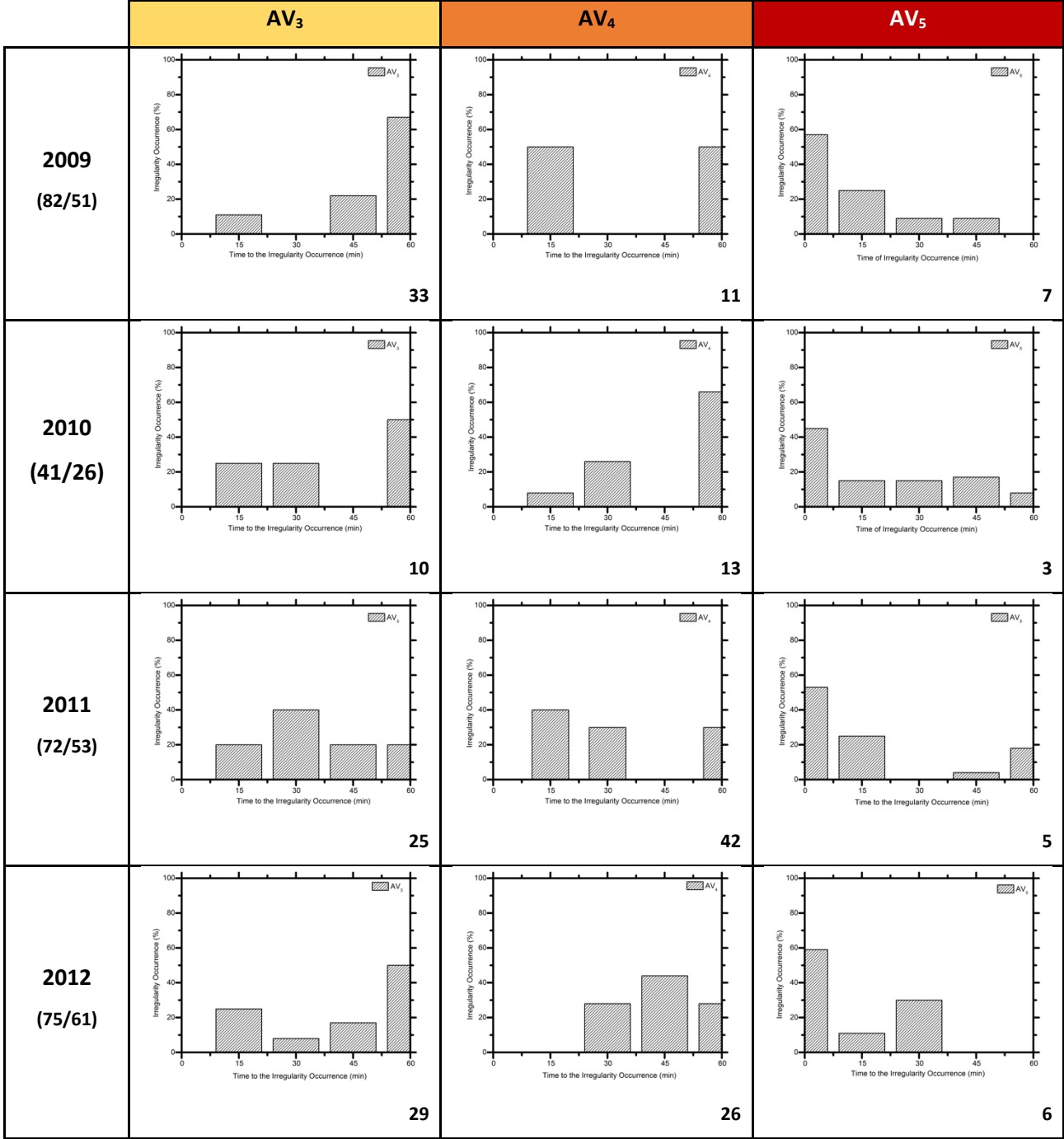

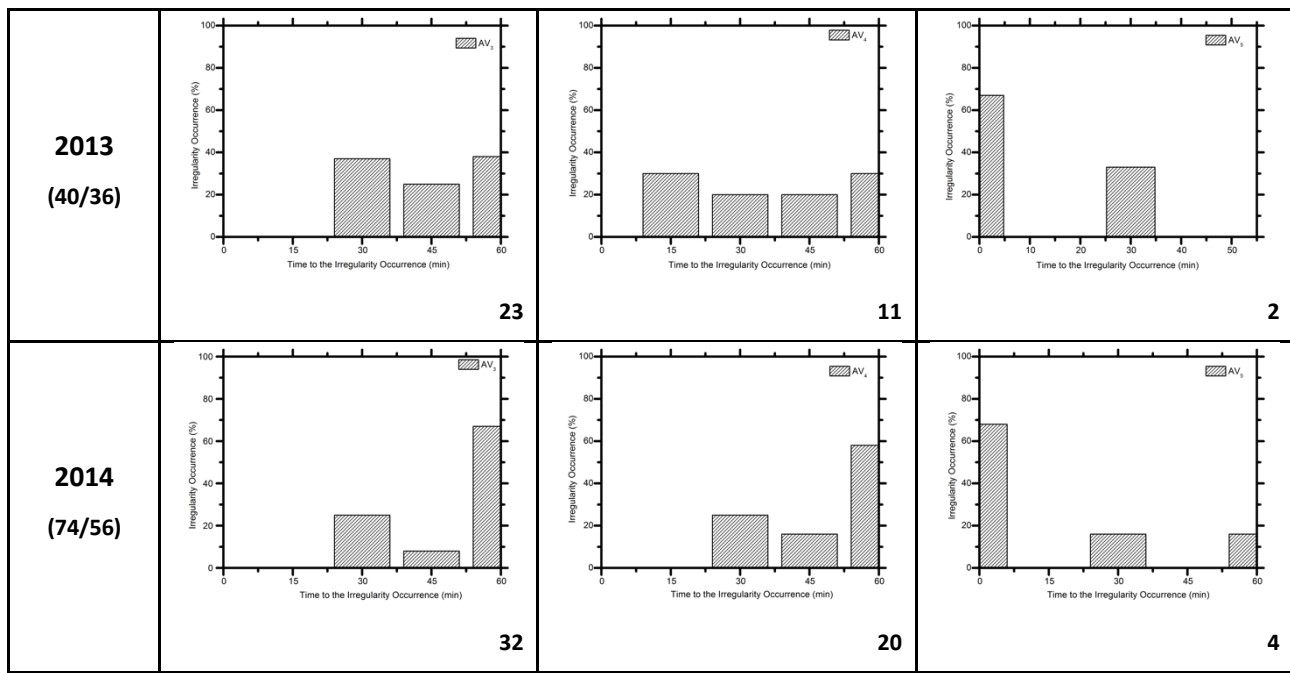

**FIGURE 6** – Time relation between the $V_{zp}$ parameter, considering AV$_3$, AV$_4$, and AV$_5$ indexes, and the irregularity observations in ionograms in the summer of 2009 until 2014 over São Luís.

We notice that when the index reached AV$_5$, there is a high occurrence of events with $\Delta t_{vi}$ equal 0 minute. This behavior is clearly observed during the ascending and maximum phases of the solar cycle, in which almost 70% of the occurrence lies inside this interval. During the minimum solar (2009 and 2010), this probability decays to ocurrences lower than 60%. This fact can be an evidence that the zonal drift reversal time and the weak zonal neutral wind magnitude can caused an delay in the irregularity occurrence during the solar minimum activity phase (Abdu et al., 1985). Despite, all these promising results, we shall recall that we are working with a 7-year interval. Thus, any definitive conclusion on the solar cyle dependence must be provide by a more comprehensive study that must confirm our results.

We included an analysis of the average $\Delta t_{vi}$ considering all years for AV$_3$ (yellow line), AV$_4$ (orange line), and AV$_5$ (red line) in Figure 7. It is possible to observe that the mean $\Delta t_{vi}$ is greater than 60 minutes, for AV$_3$ and AV$_4$. In severe events, *i.e.* AV$_5$, we had the mean $\Delta t_{vi}$ equal 0 minute. Thus, we can infer that, under AV$_3$ and AV$_4$, the elapsed time between the $V_{zp}$ peak and the irregularity occurrence is greater than 60 minutes with considerable probability (around 50%).

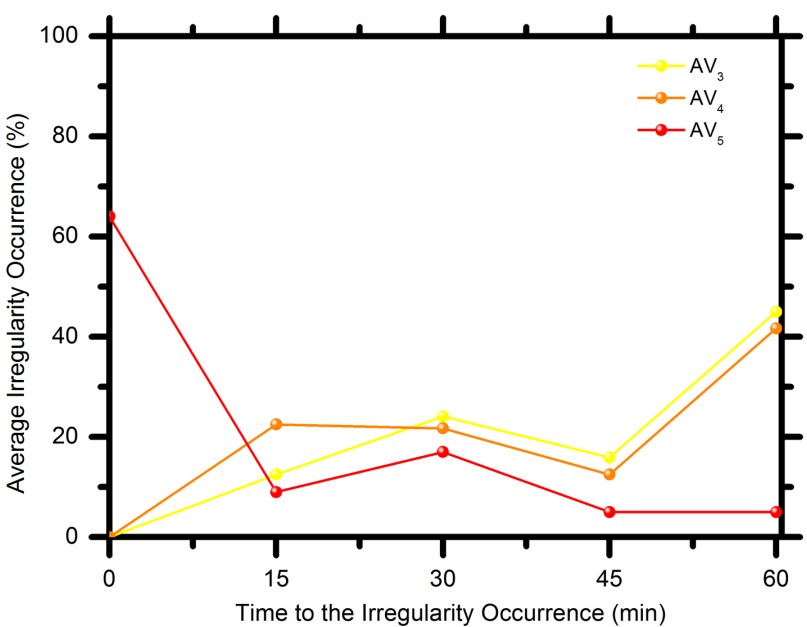

**FIGURE 7 -** Average time relation between the $V_{zp}$ parameter, considering $AV_3$ (yellow), $AV_4$ (orange), and $AV_5$ (red) indexes, and the irregularity observations in ionograms in summer during 2009 until 2014 over São Luís.

It is well known that the $\frac{E}{B}$ term in Equation 1 arises from the evening vertical drift enhancement, also known as PRE (Abdu, 2009b). Therefore, when the base of the F layer lies above 300 km, this term ($\frac{E}{B}$) is approximately equal to $V_{zp}$ (Bittencourt and Abdu, 1981; Batista et al., 1986). Furthermore, the term ($\frac{g}{\nu}$) can be enhanced due to a large $V_{zp}$, leading to an enhancement of the linear growth rate ($\gamma$). In other words, there is a clear compromise between the $V_{zp}$ and the linear growth rate of the plasma bubble. Moreover, several authors have reported the necessity to have a wave-like perturbation to trigger the Rayleigh-Taylor instability (Abdu et al., 2009b, Takahashi et al., 2018, Tsunoda, 2006).

Using data from ground-based experiments conducted during the 2005 SpreadFEx campaign in Brazil, Abdu et al. (2009c) have shown few case studies where it is possible to identify a relationship between $V_{zp}$ and plasma bubble occurrence. The authors observed that the spread-F occurs 30 minutes after the $V_{zp}$ reached values around 40 m/s. They concluded that the relationship between $V_{zp}$ and plasma bubble occurrence is a significant factor to understand the development of these irregularities. Moreover, as

mentioned before, several authors have reported the necessity to have a wave-like perturbation to trigger the Rayleigh-Taylor instability, responsible for the plasma irregularities formation in the F region (Abdu et al., 2009c; Tsunoda, 2006). Narayanan et al. (2014) studied the connection between the equatorial spread-F and satellite traces over an Indian equatorial station, Tirunelveli (8.7°N, 77.8°E, dip: 1.1°N). They used ionosonde observations with 5-min intervals from March 2008 to February 2009 during the extended solar minimum conditions. Among their results, they found an average spread-F onset delay of about 30 min to the satellite traces observation. Therefore, they believed that the satellite trace remarks may be used as a precursor of the irregularities occurrence. Additionally, they recommended validating their analysis with the $V_{zp}$ parameter, but did not perform it.

In a recent study, Sousasantos et al. (2019) developed a three-dimensional numerical model to analyze plasma bubbles structure. An input parameter is the PRE (ranging between 20 and 60 m/s) obtained using the SAMI2 model (Huba et al., 2000), wich played a central role, being a necessary condition in addition to all mechanisms that to form plasma bubbles. Thus, the authors showed that the plasma bubble structure is generated approximately 20-30 minutes after the PRE enters in the model. They attributed this delay to the parallel conductivity that may reduce the growth rate on the equatorial region. In turn, it implyes a larger times required to reach the nonlinear stage.

As presented, the relationship between $V_{zp}$ and irregularities onset time is an open important scientific issue. Based on the previous studies, both the wave-like perturbations or the parallel conductivity can modify the outset of the irregularity, being the responsible for the delay. In order to settle the key factor, a more comprehensive and specific study shall be designed.

Additionally, the ionospheric indexes found are related to the TEC or satellites measurements with the plasma irregularity occurrence (Huang et al., 2015; Nishioka et al., 2017). On the other hand, there is no ionospheric index in the available literature that related the drift velocity with the irregularity/plasma bubbles occurrences. Furthermore, this study confirms that this proposed index can be used to warn the users about the irregularity occurrences, since it was shown that under $AV_3$ and $AV_4$ there are at least 15 minutes between $V_{zp}$ peak observation and the irregularity occurrence.

Finally, as a perspective for future works, this index will be useful to study the seasonality, solar cycle, and onset time of the plasma irregularities. Notice that the temporal error is around 15 minutes,

since the ionogramas has this renge of time. Howerve, we believe that the error is not significant in terms of the spread F occurrence. Therefore, the AV index is suitable to be incorporated into the products offered by the Embrace program, and it will help in the evaluation of the phenomena impacts in the space weather environment.

## 4 Conclusions

In this study, we develop an ionospheric index, AV, based in the $V_z$ parameter in the F region. The index quantifies the time relation between $V_{zp}$ and the ionospheric irregularity occurrences that may be associated with plasma bubbles, $\Delta t_{vi}$. We analyzed two years representing the different solar flux, 2001 and 2015 to construct the AV index scale. After, we performed a climatological study of 2009 until 2014, in which to validate the AV scale.

In general, the results show that $\Delta t_{vi}$ is at least 15 minutes in both solar cycles for $AV_3$ and $AV_4$ indexes. However, when the index reached the $AV_5$, in which it is considered extreme events, we observed a high occurrence of events with $\Delta t_{vi}$ equal 0 minute (60% of the cases).

Additionally, we observed a significantly high number of irregularities in 2015 (61 cases in summer; 58 in equinoxes, and 23 in winter). We attributed this fact due to the descending phase of the solar cycle. However, although the irregularities reached $AV_5$ in 2015, its duration in ionograms was lower than that in 2001. We believe that the thermospheric winds are the main agent responsible for this behavior, since they interfere in the growth rate of Rayleigh-Taylor instability. This fact will be evaluated in greater detail in future work.

We performed a climatological study during the summer since this season is more significant in spread-F occurrences. Thus, we considered the data obtained from São Luís station of 2009 up to 2014, covering almost the ascending solar cycle. We show an irregular pattern between the highest $V_{zp}$ value and the irregularity observations. However, it is important to mention that the probability to observe $\Delta t_{vi}$ equal to or higher than 15 minutes is still very high in all phases of the solar cycle. In fact, we can infer that, under $AV_3$ and $AV_4$, the elapsed time between the $V_{zp}$ peak and the irregularity occurrence is greater than 60 minutes with very high probability.

Finally, in the previous studies the wave-like perturbations to trig the Rayleigh-Taylor instability or parallel conductivity can modify the outset of the irregularity, and which should be responsible for the delay between $V_{zp}$ and irregularities occurrence. However, more studies are needed to understand this relation. In fact, we did not find any study about the ionospheric index that related the drift velocity with the irregularity/plasma bubbles occurrences as shown in this work. Thus, the AV index is suitable to be incorporated into the products offered by the Embrace program, and it will help in the evaluation of the phenomena impacts in the space weather environment.

**Acknowledgments**

L. C.A. Resende would like to thank the National Space Science Center (NSSC), Chinese Academy of Sciences (CAS) for supporting her postdoctoral, and the CNPq/MCTIC (grant 169404/2017-0). C. M. Denardini thanks CNPq/MCTI, grant 03121/2014-9. G. A. S. Picanço would like to thank CNPq for the financial support received during his M.Sc. (grant 132252/2017-18). J. Moro would like to thank the National Space Science Center (NSSC), Chinese Academy of Sciences (CAS) for supporting his postdoctoral, and the CNPq/MCTIC (grant 429517/2018-01). D. B. thanks the CNPq fellowship (grant 301211/2019-1). C. A. O. B. Figueiredo thanks the FAPESP Postdoctoral fellowship (grant 2018/09066-8). R. P. Silva thanks the support from CNPq by grant 300329/2019-9. The authors thank DAE/INPE for kindly providing the ionospheric data.

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
