# Peer review of "On developing a new ionospheric plasma index for the Brazilian equatorial F region irregularities"

_Annales Geophysicae, 2019_

## Referee Comment (RC1) · Anonymous Referee #1 · 1 Apr 2019

This paper reports relationship between vertical drift velocity (Vz) of the F layer and spread-F occurrence. Although the authors describe that a new index is introduced, the index is defined simply as magnitude of Vz. It is well-known that the spread-F occurrence strongly depends on Vz. This reviewer does not consider that introduction of the new index is valuable. However, the results in this study show that the spread-F occurrence is delayed at least 15 min from the time of Vz when Vz is less than 60 m/s, but that there is no time delay when Vz exceeds 60 m/s. This reviewer considers that these results are most important finding in this paper. Consequently, this reviewer recommends the authors to emphasize this point more in the revised manuscript so that this paper is worth publishing in this journal.

[Figure]

Minor comments: – Describe which months are classified into summer, equinox, and winter. – Table 1: Each level of AV index is defined as vertical velocity larger than a threshold and smaller than another threshold. When the velocity is equal to the threshold, how the level is defined?

———————————————————

---

## Editor Comment (EC1) · Igo Paulino (Editor) · 1 Apr 2019

Dear Authors!

First of all, thank you for choose the Annales Geophysicae to publish the results of your research.

The idea to propose an index as a kind of warning for the occurrence of plasma bubbles sounds very good to the community. I believe that, once your algorithm is ready, it can be useful to Space Weather services.

In order to start the discussion, I have two quick questions on the methodology de-

scribed in the manuscript:

(1) Why do the authors use only high and descending phases of the solar cycle to validade the methodology?

(2) Is there a special reason for choose November, December and January as summer months? I guess December, January and February would be more representative of the summer season, are not them? The same comment can be applied to the winter months. Why do not the authors use February in their analysis?

---

## Author Comment (AC1) · 18 Apr 2019

Manuscript 'On developing a new ionospheric plasma index for the Brazilian equatorial F region irregularities' by Resende et al. submitted to the Annales Geophysicae.

We would like to thank the editor for kindly evaluating this manuscript. We have answered the questions below.

Questions:

1. Why do the authors use only high and descending phases of the solar cycle to validade the methodology?

[Figure]

Our response: The low latitude ionograms is aware of the necessity to manually checking of the ionograms for reliable results. Therefore, the criterion was to choose two whole years in different phases of the solar cycle that had a complete and revised amount of data to avoid any error in the results. So, this fact occurred in the years of 2001 (high solar activity) and 2015 (descending phase of the cycle). We added this information in page 4, line 4-6.

2. Is there a special reason for choose November, December and January as summer months? I guess December, January and February would be more representative of the summer season, are not them? The same comment can be applied to the winter months. Why do not the authors use February in their analysis?

Our response: We chose the months when the irregularities were observed almost every night in the ionograms. In other words, since plasma bubbles occur between September and March, we eliminate the two months before and after this analysis. The same criterion was used in the winter season. For better understanding, we add this information in the text (line 10-13, page 4).

Finally, we would like to take this opportunity to thank the editor for kindly evaluating our paper helping to greatly improve its quality.

Please also note the supplement to this comment:
https://www.ann-geophys-discuss.net/angeo-2019-42/angeo-2019-42-AC1-supplement.pdf

---

## Author Comment (AC2) · 18 Apr 2019

Responses to the Comment and/or Suggestions from Referee 1

Manuscript 'On developing a new ionospheric plasma index for the Brazilian equatorial F region irregularities' by Resende et al. submitted to the Annales Geophysicae.

We would like to acknowledge the comments given by the referee. We have carried out a revision of the manuscript taking into account all the referee's comments and suggestions.

Comment:

1. This paper reports relationship between vertical drift velocity (Vz) of the F layer and spread-F occurrence. Although the authors describe that a new index is introduced, the index is defined simply as magnitude of Vz. It is well-known that the spread-F occurrence strongly depends on Vz. This reviewer does not consider that introduction of the new index is valuable. However, the results in this study show that the spread-F occurrence is delayed at least 15 min from the time of Vz when Vz is less than 60 m/s, but that there is no time delay when Vz exceeds 60 m/s. This reviewer considers that these results are most important finding in this paper. Consequently, this reviewer recommends the authors to emphasize this point more in the revised manuscript so that this paper is worth publishing in this journal. Our response: Thanks to the reviewer for the suggestion. The exact explanation about the time difference between Vzp and irregularities occurrence is an open question. However, we added some assumptions based on previous works (page 16, lines 4-19).

Minor comments:

1. Describe which months are classified into summer, equinox, and winter. Our response: In page 4 (lines 9-11) is described this information.

2. Table 1: Each level of AV index is defined as vertical velocity larger than a threshold and smaller than another threshold. When the velocity is equal to the threshold, how the level is defined? Our response: Thanks to the reviewer for pointed this fact. We included the symbol equals where we consider the threshold in table.

Finally, we would like to take this opportunity to thank the reviewer for kindly evaluating our paper helping to greatly improve its quality.

Please also note the supplement to this comment:
https://www.ann-geophys-discuss.net/angeo-2019-42/angeo-2019-42-AC2-supplement.pdf

2019.

**Supplement:**

[revised manuscript text omitted]
, 2006). However, the exact explanation about the time difference between $V_{zp}$ and irregularities occurrence is an open question. Based on previous works cited before, we believed that the wave-like perturbations modified the terms in the Equation 1 may cause this difference. Nevertheless, this subject is out of the scope of this paper. Here, the focus of our discussion is on development of a new ionospheric index.

Finally, to the best knowledge of the authors, there is no ionospheric index in the available literature that related the drift velocity with the irregularity/plasma bubbles occurrences. The ionospheric indexes found are related to the TEC or satellites measurements with the plasma irregularity occurrence (Huang et al., 2015; Nishioka et al., 2017). Additionally, this study confirms that this proposed index can be used to warn the users about the irregularity occurrences, since it was shown that under $AV_3$ and $AV_4$ there is at least 15 minutes between $V_{zp}$ peak observation and the irregularity occurrence, with a high probability of the $\Delta t_{vi}$ to be greater than 60 minutes. Lastly, the AV index is suitable to be incorporated into the products offered by the Embrace program, and it will help in the evaluation of the phenomena impacts in the Space Weather environment.

**4 Conclusions**

In this study, we develop an ionospheric index, AV, based in the $V_z$ parameter in the F region. The index quantifies the time relation between $V_{zp}$ and the ionospheric irregularity occurrences that may

be associated with plasma bubbles, $\Delta t_{vi}$. We analyzed two years representing the different solar flux, 2001 and 2015 to construct the AV index scale. After, we performed a climatological study of 2009 until 2014, in which to validate the AV scale.

In general, the results show that $\Delta t_{vi}$ is at least 15 minutes in both solar cycles for $AV_3$ and $AV_4$ indexes. However, when the index reached the $AV_5$, in which it is considered extreme events, we observed a high occurrence of events with $\Delta t_{vi}$ equal 0 minute (60% of the cases).

Additionally, we observed a significantly high number of irregularities in 2015 (61 cases in summer; 58 in equinoxes, and 23 in winter). We attributed this fact due to the descending phase of the solar cycle. However, although the irregularities reached $AV_5$ in 2015, its duration in ionograms was lower than that in 2001. We believe that the thermospheric winds is the responsible for this behavior, since they interfere in the growth rate of Rayleigh-Taylor instability. This fact will be evaluated in greater detail in future work.

We performed a climatological study during the summer since this season is more significant in spread-F occurrences. Thus, we considered the data obtained from São Luís station of 2009 up to 2014, covering almost the solar cycle. We show an irregular pattern between the highest $V_{zp}$ value and the irregularity observations. However, it is important to mention that the probability to observe $\Delta t_{vi}$ equal or higher than 15 minutes is still very high in all phases of the solar cycle. In fact, we can infer that, under $AV_3$ and $AV_4$, the elapsed time between the $V_{zp}$ peak and the irregularity occurrence is greater than 60 minutes with very high probability.

Finally, we believe that this proposed index can be used to warn the users about the irregularity occurrences, since it was shown that under $AV_3$ and $AV_4$ there is at least 15 minutes between $V_{zp}$ peak observation and the irregularity occurrence, with a high probability of the $\Delta t_{vi}$ to be greater than 60 minutes. And, we did not found 
[revised manuscript text omitted]

---

## Referee Comment (RC2) · Anonymous Referee #2 · 26 Apr 2019

The paper proposed by Araujo et al. introduces a new ionospheric index for Brazilian equatorial region using the Vz, calculated from digissonde h'F values, and tested mainly in summer conditions for the solar cycle 24. They show the relationship of Vz with the start time occurrence of irregularities, and stablishing a AV(1-5) index.

This paper is very interesting from my point of view, and could be published after some technical revision.

What's the difference between AV1 and AV2? Since it's not explained in all the paper. Also is not shown the statistics of AV1 and AV2, probably it could be important for Space Weather issues.

Also, I couldn't find some relation on the construction cof the AV index with the solar cycle. Probably if you (authors) could add a factor into your equation that could show the solar cycle index. Is properly understood that solar cycle affect the irregularity formation process, as well the season, so, when you are in solar maximum PRE (Vz) is well related with irregularities formation (as you mentioned in pag 8), however during solar minumum PRE (Vz) is not longer the best factor to initiate the irregularities. Also the time that the irregularities starts is different from solar maximum to solar minimum.

Why the statistics of other seasons are not shown in the paper? Do they show same behavior as Figure6? I found very interesting Figure 6 since it shows the relation of irregularities with time along 6 years data (from minimum to ascending fase of solar cycle). I believe that in order to develop an index you have to show the same statistic for the other 3 seasons (spring Equinox, fall equinox, and Winter solstice).

This AV index seems to be working properly for summer solstice, however is not possible to conclude, from the paper, that is effective for other seasons.

What happens if start of irregularities are observed at midnight and/or post-midnight hours? This AV index does not represent them.

Minor comments/revisions:

1. Pag. 1, Abstract, Line 22: "ionogramas" to "ionograms" .

2. Pag. 4, 3.1 AV Index Scale, Line 16: "plasma bubbles is more probab..." to "... are more pro...".

3. Pag 5, Line 10: Since Vzp = 53 so AV4 and not AV3.

4. Table 2: Below Vzp shouldn't be 53?

5. Pag 9, Line 4: "... in Figure 2, where..." change for Figure 3.

6. Pag 10, Lines 20 and 21: Please check the percentage numbers. For example: "Among those, 15% had..." change to "... 10%...". I believe that is related to the Figure

4, AV4, 2001.

7. Pag 12, Line 16: complete the paragraph "... of days that...".

---

## Editor Comment (EC2) · Igo Paulino (Editor) · 8 May 2019

Dear Authors!

Thank you for the quick reply to the reviewer comments. I guess that your paper has significantly been improved. However, I have read the sentence on page 16 lines 4-19, which it was added to address the major concern from reviewer #1, and I guess it was not solved properly at all.

Before considere your paper for publication, please, revise the paper again and improve your discussion. I agree with the reviewer #1 that the proposal index will be useful only

when the Vz is very low, i.e., during the months when we have the low occurrence rate of EPBs.

Best regards,

Igo
* * *

---

## Editor Comment (EC3) · Igo Paulino (Editor) · 8 May 2019

Based on the comments from reviewer #2, please, try to fix the minor points, e.g., the meaning of AV1 and AV2, etc.

I agree with the reviewer #2 that the solar cycle must be included elsewhere in your modelling. If the PRE depends on the solar cycle, the occurrence of EPB must depend on solar activity as well, does not it?

The other points, you have explained in AC1 document.

[Figure]

Best regards,

igo
* * *

---

## Author Comment (AC3) · 29 May 2019

Manuscript 'On developing a new ionospheric plasma index for the Brazilian equatorial F region irregularities' by Resende et al. submitted to the Annales Geophysicae.

Comment:

  Thank you for the quick reply to the reviewer comments. I guess that your paper has significantly been improved. However, I have read the sentence on page 16 lines 4-19, which it was added to address the major concern from reviewer #1, and I guess it was not solved properly at all. Before consider your paper for publication, please,

revise the paper again and improve your discussion. I agree with the reviewer #1 that the proposal index will be useful only when the Vz is very low, i.e., during the months when we have the low occurrence rate of EPBs.

We would like to acknowledge the comments given by the editor. We improved the discussion in the part that Reviewer 1 had suggested. The Reviewer 1 mentioned that the Vzp and their relationship with the plasma bubbles is well known, and the most interesting factor to explore was the delay between the Vzp and the spread-F. Although we did not find the real reason why that delay occurs, we discussed this fact in terms of irregularity formation and their dynamics. In addition, we compare our results with very few studies discussing this topic. Additionally, the purpose of this work is to show that it is possible to have an index that related the ionospheric parameters with the irregularity occurrences. This index can be incorporated into the products offered by the Space Weather program to facilitate the users to identify the occurrence the irregularities by only colour scale. In this work, we would just like to show that to typical values of the Vzp parameter, the irregularity in ionograms occurs after at least 15 minutes. So, as future studies, we believed that this index will be useful to study the plasma irregularities seasonality, solar cycle, and plasma irregularities onset.

Finally, we would like to take this opportunity to thank the editor for kindly evaluating our paper helping to greatly improve its quality.

Please also note the supplement to this comment:
https://www.ann-geophys-discuss.net/angeo-2019-42/angeo-2019-42-AC3-supplement.pdf
* * *

---

## Author Comment (AC4) · 29 May 2019

Manuscript 'On developing a new ionospheric plasma index for the Brazilian equatorial F region irregularities' by Resende et al. submitted to the Annales Geophysicae.

The paper proposed by Araujo et al. introduces a new ionospheric index for Brazilian equatorial region using the Vz, calculated from digissonde h'F values, and tested mainly in summer conditions for the solar cycle 24. They show the relationship of Vz with the start time occurrence of irregularities, and establishing a AV(1-5) index. This paper is very interesting from my point of view, and could be published after some technical revision.

We would like to acknowledge the comments given by the referee. We have carried out a revision of the manuscript taking into account all the referee's comments and suggestions.

Comment:

1. What's the difference between AV1 and AV2? Since it's not explained in all the paper. Also is not shown the statistics of AV1 and AV2, probably it could be important for Space Weather issues. Our response: Thanks to the reviewer for this point. To construct our scale, we observe that the irregularities occur after Vzp reaches 40 m/s in general. As for product works of Space Weather have a greater interest in general cases, we consider the values smaller than 40 m/s as conditions of the irregularity will not appear (AV1) or will rarely appear (AV2). We added this information in page 5, lines 10-13.

2. Also, I couldn't find some relation on the construction of the AV index with the solar cycle. Probably if you (authors) could add a factor into your equation that could show the solar cycle index. Is properly understood that solar cycle affect the irregularity formation process, as well the season, so, when you are in solar maximum PRE (Vz) is well related with irregularities formation (as you mentioned in pag 8), however during solar minumum PRE (Vz) is not longer the best factor to initiate the irregularities. Also the time that the irregularities starts is different from solar maximum to solar minimum. Our response: The reviewer is right when he/she emphasizes the relationship between plasma irregularities and solar activity. We observe a relation to the instant in which Vzp increases (using Vzp equation) with the time that the irregularity is identified in AV3, AV4, and AV5 indexes in all solar cycle phase studied. However, we observe some differences according to the phase of the solar cycle. One of them is the time of duration of the irregularity (Figure 3 discussion). Also, we noted the AV5 index seems to have a delay of the irregularity occurrence time. We added this part on page 15 (lines 5-10). More detailed studies using this index and the irregularity observations regarding the phase of the solar cycle intended to be analysed in the future.

3. Why the statistics of other seasons are not shown in the paper? Do they show same behavior as Figure 6? I found very interesting Figure 6 since it shows the relation of irregularities with time along 6 years data (from minimum to ascending fase of solar cycle). I believe that in order to develop an index you have to show the same statistic for the other 3 seasons (spring Equinox, fall equinox, and Winter solstice). This AV index seems to be working properly for summer solstice, however is not possible to conclude, from the paper, that is effective for other seasons. Our response: As mentioned on page 12, we observed only a few cases of spread-F in winter and equinoxes. Also, as pointed out by reviewer 1, it is already well known that the irregularities related to plasma bubbles are more frequent in summer. Therefore, we opted to show only the summer season in order not to be a very long article. For exemplify, we show the winter statistics here. In fact, we observe that the probability to observe the time that the irregularity starts to appear in ionogram after the Vzp intensification is equal to or higher than 30 minutes to AV3 and AV4 indexes. Thus, notice that in the few cases that the irregularity occurred, the index has the same behaviour than summer.

4. What happens if start of irregularities are observed at midnight and/or post-midnight hours? This AV index does not represent them. Our response: Thank you to the reviewer for this important question. The irregularities that happen at this time (midnight and/or post-midnight hours) are counted in an hour or more in our statistics (>60 minutes). We added this information (page 8, lines 5-10; page13 and 14).

Minor comments:

1. Pag. 1, Abstract, Line 22: "ionogramas" to "ionograms". Our response: We correct it accordingly. Thank you.

2. Pag. 4, 3.1 AV Index Scale, Line 16: "plasma bubbles is more probab..." to "... are more pro...". Our response: We correct it accordingly.

3. Pag 5, Line 10: Since Vzp = 53 so AV4 and not AV3/Table 2: Below Vzp shouldn't be 53? Our response: Sorry, the Vzp was 43 m/s. We correct it accordingly. 4. Pag

9, Line 4: "... in Figure 2, where..." change for Figure 3. Our response: We correct it accordingly. 5. Pag 10, Lines 20 and 21: Please check the percentage numbers. For example: "Among those, 15% had..." change to "... 10%...". I believe that is related to the Figure 4, AV4, 2001. Our response: We correct it accordingly. 6. Pag 12, Line 16: complete the paragraph "... of days that...". Our response: We correct it accordingly.

Finally, we would like to take this opportunity to thank the reviewer for kindly evaluating our paper helping to greatly improve its quality.

Please also note the supplement to this comment:
https://www.ann-geophys-discuss.net/angeo-2019-42/angeo-2019-42-AC4-supplement.pdf

---

## Author Comment (AC5) · 29 May 2019

Manuscript 'On developing a new ionospheric plasma index for the Brazilian equatorial F region irregularities' by Resende et al. submitted to the Annales Geophysicae.

Comment:

• Based on the comments from reviewer #2, please, try to fix the minor points, e.g., the meaning of AV1 and AV2, etc.. Our response: We have carried out a revision of the manuscript taking into account all the referee's comments. And, we added the AV1 and AV2 information in lines 10-13.

• I agree with the reviewer #2 that the solar cycle must be included elsewhere in your modelling. If the PRE depends on the solar cycle, the occurrence of EPB must depend on solar activity as well, does not it? • Our response: As we explained to the reviewer, we observe a relation to the instant in which Vzp increases (using Vzp equation) with the time that the irregularity is identified in AV3, AV4, and AV5 indexes in all solar cycle phase studied. We only observe some differences in relation to the phase of the solar cycle, in which we added in our discussion (page 15, lines 5-10).

Finally, we would like to take this opportunity to thank the editor for kindly evaluating our paper helping to greatly improve its quality.

Please also note the supplement to this comment:
https://www.ann-geophys-discuss.net/angeo-2019-42/angeo-2019-42-AC5-supplement.pdf

---

## Author Comment (AC6) · 29 May 2019

The comment was uploaded in the form of a supplement:
https://www.ann-geophys-discuss.net/angeo-2019-42/angeo-2019-42-AC6-supplement.pdf

---

## Author Comment (AC7) · 29 May 2019

**On developing a new ionospheric plasma index for the Brazilian equatorial F region irregularities**

5 Laysa Cristina Araujo Resende1,2, Clezio Marcos Denardini1, Giorgio Arlan Silva Picanço1, Juliano Moro2,3, Diego Barros1, Cosme Alexandre Oliveira Barros Figueiredo1, Régia Pereira Silva1.

[revised manuscript text omitted]

---

## Author Response (AR2)

**Responses to the Comment and/or Suggestions from Referee 1**

Manuscript 'On developing a new ionospheric plasma index for the Brazilian equatorial F region irregularities' by Resende et al. submitted to the Annales Geophysicae.

*We would like to acknowledge the comments given by the referee. We have carried out a revision of the manuscript taking into account all the referee's comments and suggestions.*

**Comment:**

1. The authors have investigated time delay of the peak of the upward velocity of the F layer movement and onset of the spread F. This reviewer believes that it is worth studying, but the authors need to mention the followings. The observations carried out in this study is the observation just above the ionosonde. By the only one point observation, it is impossible to distinguish between temporal and spatial variations of the spread F or plasma bubble. In most cases, plasma bubbles occur around sunset terminator and move eastward at the same velocity as the ambient plasma in the ionosphere. By the observation at a single point, it is impossible to identify where and when the plasma bubble is generated. For an example, we consider the case that spread F is observed by an ionosonde 1 hour after the upward velocity of the F layer movement reaches a peak. Plasma bubble may be generated above the ionosonde 1 hour after the peak of the upward velocity of the F layer. The other possibility that the plasma bubble is generated at west of the ionosonde site, moves eastward, and reaches the ionosonde site 1 hour after the peak of the upward velocity of the F layer. Satellite trace in the ionogram arises from radio wave reflection from the oblique direction, indicating plasma bubble does not exist just above the ionosonde, but apart from the ionosonde.

*Our response:*

*Thanks to the reviewer for highlight this point. However, the digisonde can make vertical and oblique soundings, which allow us to study a considerable portion of the ionosphere. The digital beam forming is done taking four complex amplitudes observed in a particular Doppler line of the spectrum on four receive antennas and forming seven beams shown in the figure below. There is one bam overhead at 0° zenith angle and six oblique beams cantered at North and South directions and each 60° in between (Reinisch et al., 2009, doi: 10.1029/2008RS004115).*

[Figure]

*Figure 1. Seven beams for the angle of arrival measurd in ionogram mode. Source: Technical Manual Operation and Maintenance of Digisonde.*

*A spatial range of ~3480 km is obtained at 250 km altitude. Therefore, the digisonde can make measurements up to ~3480 km away from the point just above it. The ordinary and extraordinary traces as well as the vertical and oblique soundings can be identified by the colours in the ionograms. Also, the digisonde can monitor the ionosphere continuality with a temporal resolution of 10 minutes. So, we can monitor the ionosphere since the pre-reversion peak up to spread f onset time with good precision (Abdu et al., 2009; doi: 10.5194/angeo-27-2607-2009). Moreover, the error is not significant in terms of the spread F occurrence, as the main purpose for the Space Weather program. Finally, we understand that satellite traces are necessary precursor to the occurrence of an spread F trace on the ionograms as point out by Cabrera et al. 2010 (doi: 10.5194/angeo-28-1133-2010 ). However, the onset time of the spread f was considered when the digisonde echoes were clear diffuse. In order to clarify these points, we include some information about the temporal error in this manuscript (page 18).*

2. P. 11, l. 1: "U^P" is not zonal wind. It is the Pedersen conductivity-weighted neutral wind perpendicular to the magnetic field in the magnetic meridian plane.

*Our response:*
*We modify this part accordingly.*

3. P. 14, ll. 15-17, "In addition, the formation of these irregularities ...": It is better to change to "the post-midnight plasma bubbles could be caused by atmospheric gravity wave seeding of the Rayleigh-Taylor instability and/or increase in growth rate of the Rayleigh-Taylor instability due to the F-layer uplift, .

*Our response:*
*We modify this part accordingly.*

4. P. 14, ll. 17-18, "In both cases, there is a decrease...":
This explanation of applicable to only latter case, that is the increase in growth rate of the Rayleigh-Taylor instability due to the F-layer uplift. Gravity wave does affect the ion-neutral collision frequency. On the other hand, when the F-layer is uplifted, the F layer exists at higher altitude, where the ion-neutral collision frequency is low.

*Our response:*
*The reviewer is right. We remove this last phrase to avoid some confusion in the text.*

**Minor comments:**

1. P. 14, l. 16, "an abrupt uplift": "abrupt" is not needed. Only "uplift of the F layer" is.

*Our response:*
   *Done.*

2. P. 14, l. 17, "In both cases, ...": When the F layer is uplifted

*Our response:*
   *This phrase was removed in this new version of Manuscript.*

3. P. 16, l. 8: what is "um"?

*Our response:*
   *We modify this part accordingly.*

4. P. 17, l. 13: "shown" --> "have shown"

*Our response:*
   *Done.*

   *Thanks to the reviewer for pointed this fact. We included the symbol equals where we consider the threshold in table.*

5. It is better to use "December solstice" and "June solstice instead of "summer" and "winter" because plasma bubbles occur around equator and exist simultaneously in both northern and southern hemispheres.

*Our response:*
   *In this case, we prefer to use the summer and winter, since the magnetic equator is located in the Brazilian sector in the southern hemisphere.*

6. Reference list: Change the order of the references to "Barros et al., 2018", "Basu et al. 1996", and "Batista et al. 1986 and 1996".

*Our response:*
   *Done.*

   *Finally, we would like to take this opportunity to thank the reviewer for kindly evaluating our paper helping to greatly improve its quality.*